# Association between household food security and socioeconomic status with paternal smoking: Findings from SEANUTS II Malaysia

Giin Shang Yeo[1], Shoo Thien Lee[1,2], Kuan Chiet Teh[1], Jyh Eiin Wong[1], Wan Siti Fatimah Wan Abdul Rahman[1], Nurul Hasanah Hasmuni Chew[1], Caryn Mei Hsien Chan[1], Nur Zakiah Mohd Saat[1], Nik Shanita Safii[1], Siti Balkis Budin[1], Swee Fong Tang[3], A. Karim Norimah[1], Ilse Khouw[4], Lei Hum Wee[1,5]*, Bee Koon Poh[1]*, on behalf of the SEANUTS II Malaysia Study Group¶

1 Faculty of Health Sciences, Universiti Kebangsaan Malaysia, Kuala Lumpur, Malaysia, 2 Faculty of Health and Life Sciences, Management and Science University, Shah Alam, Malaysia, 3 Department of Paediatrics, Specialist Children's Hospital, Universiti Kebangsaan Malaysia, Kuala Lumpur, Malaysia, 4 FrieslandCampina, Amersfoort, the Netherlands, 5 Faculty of Health and Medical Sciences, School of Medicine, Taylor's University, Selangor, Malaysia

¶ Membership of the SEANUTS II Malaysia Study Group is provided in Acknowledgments.
* weeleihum@ukm.edu.my, leihum.wee@taylors.edu.my (LHW); pbkoon@ukm.edu.my (BKP)

## Abstract

Individuals burdened with socioeconomic disadvantages, including food insecurity, are often exposed to circumstances that may increase the likelihood of smoking, which may in turn affect the respiratory health of others in the household, particularly children. Yet, the role of these factors in perpetuating tobacco use within households remains underexplored. Therefore, this study attempts to determine the association between household food security and socioeconomic status with paternal smoking in Malaysia. This study utilised data from the South East Asian Nutrition Surveys (SEANUTS II) Malaysia, which involved 2,687 children aged 0.5–12.9 years, recruited through multistage cluster sampling. Although the study recruited children as participants, the present analysis draws on paternal and household information, including food security, socioeconomic status, and paternal smoking, as reported through parent/guardian-reported questionnaires. Multivariate binary logistic regression was used to determine factors associated with paternal smoking habits. Overall, forty-three percent of fathers were reported as smokers. Younger fathers (aged <30.0 years) had twice the odds of smoking [aOR: 2.032 (95%CI 1.301–3.173)] compared to their older counterparts (aged >49.9 years). Fathers from low socioeconomic status households, particularly, those with extremely low household income [aOR: 2.606 (95%CI 1.711–3.968)] and without formal education/completed primary education [aOR: 2.604 (95%CI 1.710–3.966)], had double the odds of smoking than their counterparts. Self-employed/other occupation fathers [aOR: 1.343 (95%CI 1.056–1.709)] and families from food-insecure households [aOR: 1.251 (95%CI 1.053–1.487)] were

**Data availability statement:** The data generated or analysed during this study are not publicly available, as participant consent for data sharing was not obtained. Although direct identifiers have been removed, the data are potentially classified as pseudonymised under the European General Data Privacy Regulations, with a possible risk of re-identification. However, the data are available upon reasonable request and with permission from project funder via official organisation email (pengerusi.pkk@ukm.edu.my).

**Funding:** This study was conducted as part of the South East Asian Nutrition Surveys II (SEANUTS II) (NN-2018-159) and was funded by FrieslandCampina, Amersfoort, The Netherlands. The funder provided support in the form of salaries for author Ilse Khouw [IK], but did not have any additional role in study design, data collection and analysis, decision to publish, or preparation of the manuscript. The specific roles of the author are articulated in the 'author contributions' section.

**Competing interests:** Ilse Khouw is an employee of FrieslandCampina. All other authors declare no competing interest.

found to have higher odds of smoking. In conclusion, household food insecurity and socioeconomic status, including age, occupation, household income, and education level were significantly associated with paternal smoking status. Authorities should customise interventions to mitigate socioeconomic barriers and create equitable access to quit-smoking resources. This is important for lowering smoking prevalence and safeguarding children's health.

## Introduction

Cigarette smoking continues to prevail amongst individuals aged 15–54 years, with approximately 1.14 billion smokers worldwide and over 8 million deaths annually, including 1.3 million non-smokers from secondhand smoke (SHS) [1]. In Southeast Asia, about 18% of tobacco-related deaths in 2019 are attributed to SHS exposure, underscoring its significant impact [2]. Consequently, both active and passive smoking pose a major public health challenge worldwide, including Malaysia. The 2023 National Health and Morbidity Survey (NHMS) reported that 19% of Malaysian adults smoke and 5% use e-cigarettes [3]. Another local study also found that half of primary schoolchildren (52.9%) lived with at least one smoker [4], placing children at higher risk of developing respiratory-related sequelae due to their faster respiratory rates and narrower airways. Additionally, both SHS and thirdhand smoke (THS) exposure can harm multiple organs and trigger inflammatory processes in the lungs [5].

Despite ongoing tobacco control initiatives, many smokers turn to e-cigarettes in hopes of quitting or reducing cigarette use; however, nicotine-containing liquids remain addictive and can reduce the likelihood of successful cessation [6]. Acknowledging that smoking is frequently adopted as a coping strategy under psychological stress [7]. Parents, especially sole breadwinners, could experience considerable emotional burden when unable to meet their family's needs. This burden is further intensified by persistent socioeconomic disadvantages, particularly widening income disparities in Malaysia [8], where household income growth lags behind inflation. Lower-income households are disproportionately affected, facing higher inflation due to greater spending on food and necessities [9]. As a result, those families struggle to allocate sufficient resources for adequate nutrition, with this financial strain often translating into household food insecurity that further amplifies psychological stress [10]. A systematic review on Malaysian population also emphasised the unexpectedly high prevalence of household food insecurity among adolescents, adults, low-income households, and welfare-recipient households [11].

While the health risks of smoking and SHS exposure are well-established, less attention has been given to the socioeconomic context that perpetuates tobacco use within households. Thus, comprehensively understanding the interplay between household food insecurity, socioeconomic conditions, and smoking habits is crucial, as these factors may exert broader implications for household well-being and children's development. Besides, addressing these interconnected issues can potentially foster the development of healthy lifestyles among younger generations, aligning with the attainment of

the United Nation's Sustainable Development Goals 2030 [12]. Hence, this study aims to investigate the association between household food security, socioeconomic contexts, with paternal smoking habits using a nationwide dataset of Malaysian children, with the aim of informing targeted interventions that address both tobacco control and household vulnerabilities.

## Methods

### Study design

A dataset from the nationwide cross-sectional survey, South East Asian Nutrition Surveys II (SEANUTS II) Malaysia, was analysed. A total of 2,989 children aged 0.5–12.9 years were recruited between May 2019 and March 2020 using a multistage cluster sampling approach across four regions of Peninsular Malaysia, namely Central, Northern, East Coast, and Southern regions. In the first stage, the Department of Statistics Malaysia (DOSM) randomly selected two districts (one urban, one rural) from different states within each region. In the second stage, DOSM provided a list of enumeration blocks within these districts. Children and institutions located within a 5 km radius of the selected blocks were eligible, with recruitment conducted via home visits (0.5–6 years) and schools (7–12 years), with additional data collection conducted in preschools and nurseries for younger children. The detailed SEANUTS II study design and sampling protocol have been described in the published articles [13,14].

All human-involved protocols were approved by the Research Ethics Committee of Universiti Kebangsaan Malaysia (Code: JEP-2018–569). Additionally, SEANUTS II Malaysia was registered with the Dutch Trial Registry (Code: NL7975) and the Guidelines of the Declaration of Helsinki were referred to in designing the study procedures. Before the commencement of data collection, permissions to conduct the project were obtained from the Ministry of Education, relevant state education departments, and other involved authority departments. Written informed consent was obtained from the parents/guardians, and verbal assent was obtained from the children.

### Socioeconomic status and paternal smoking habit

Socioeconomic status (SES) and paternal smoking information were collected from the parents/ guardians-reported questionnaires. The total household monthly income was categorised into four groups [8]: (1) extremely low (≤ MYR 2,208); (2) low (MYR 2,209-4,849); (3) middle (MYR 4,850–10,959) and (4) high (> MYR 10,959) [USD 1 = MYR 4.13 (as at 29th November 2025)]. Similarly, paternal education level was classified into three main groups: (1) no formal education/primary education; (2) secondary education; and (3) tertiary education. Paternal occupation was categorised according to profession and education requirements [15], which are (1) Professionals/Technicians and associate professionals/Managerial; (2) Clerical/Service and sales staff; (3) Agricultural, forestry and fishery/Craft and trades/Plant and machine/Elementary workers; (4) Armed-forces occupations; (5) Self-employed/Other occupations; and (6) Not working/Retiree. The groups based on paternal age were: (1) below 30.0 years; (2) 30.0–39.9 years; (3) 40.0–49.9 years; and (4) above 49.9 years.

Paternal smoking status was assessed through questions on current cigarette and vape use, based on CDC and WHO surveillance definitions [16]. Fathers were classified as smokers or vapers if they reported using cigarettes or vapes "every day" or "some days" at the time of the survey [16]. Fathers who reported being smokers or vapers were further asked about their tobacco/e-cigarette use, including frequency (e.g., number of cigarettes per day, daily vaping frequency) and intensity (e.g., preferred nicotine concentration in vape liquids). Additional questions captured whether smoking or vaping occurred at home and the time to first cigarette or vape after waking, enabling a detailed assessment of paternal smoking addiction levels. For analysis, paternal smoking status was dichotomised as smoker (current smoker, current vaper, or both) and non-smoker.

### Household food security status

The Radimer/Cornell Hunger and Food Insecurity instrument was utilised [17]. This 10-item questionnaire answered by parents or guardians, which reflects the household food security status with three options provided: not true (negative

answer); sometimes true, and always true (positive answer). Household food security was scored and classified into 4 groups with increasing severity, namely: household food secure (negative answers for all items); household food insecure (positive answer(s) to the first four items); individual food insecure (positive answer(s) to the next 3 items); and child hunger (positive answer(s) to the last 3 items). The questionnaire was translated and validated in a local study [18].

## Data analysis

Complete case analysis was performed, excluding incomplete and invalid data (n = 302), using IBM SPSS Statistics for Windows version 22.0 (IBM Corp, Armork, New York, USA). Mean and standard deviation (SD) were reported for continuous SES variables. The proportions in paternal smoking by SES and household food security level were presented as percentages. The association between household food security levels and socioeconomic groups with paternal smoking categories, was analysed using the Pearson's Chi-square test. Independent variables with $p < 0.05$ in the Chi-square test were further examined in univariate models. Variables that remained significant were included in the final logistic regression model, while those non-significant variables were treated as confounders in the adjusted model.

Before conducting the regression, independent variables were coded using dummy variables, and multicollinearity was assessed by examining variance inflation factors (VIF) and tolerance values in the multivariate linear regressions. All independent variables in the final model had VIF < 5 and tolerance >0.1, confirming no multicollinearity. Subsequently, multivariate binary logistic regression was performed to estimate the odds ratio (OR) of paternal smoking status in relation to household food security and SES-related variables, using non-smoking fathers as the reference group. Two models were presented: Model 1: crude, and Model 2: adjusted with ethnicity and area of residence. The significance level was set as $p < 0.05$ in all statistical analyses.

## Results

Table 1 presents the sample distribution according to household food security and SES. Data on 2,687 children aged 0.5 to 12.9 years and information on their fathers were included in the analysis. Most of the fathers (85.5%) were within the age groups of 30–49 years, namely 30.0–39.9 years (45.9%) and 40.0–49.9 years (39.6%). More than one-third of the fathers (35.8%) worked as Professionals/Technicians and associate professionals/Managerial, followed by Self-employed/Other Occupations in 24.2% of the households. Around one-fourth of children (24.5%) came from households with an income below the poverty line (≤MYR 2,208), and only 8.7% of households were categorised in the high-income group (>MYR 10,959). Two out of five fathers (42.2%) completed tertiary education level and only 5.0% either had no formal education or completed primary education. In terms of household food security, three-fifths of households (60.6%) reported having no issues. The distribution of families with household food insecurity, individual food insecurity, and child hunger was 17.7%, 6.3%, and 15.4%, respectively.

Results on paternal smoking characteristics and related habits are depicted in Table 2. Forty-three percent of fathers smoked; most (68.2%) were conventional cigarette smokers, with 10.1% using e-cigarettes, 13.0% using both, and 8.7% classified as missing data. Approximately one-third of fathers (36.3%) reported that they did not smoke at home. By exploring the smoking addiction level, nearly two-fifths of fathers (37.9%) smoked their first cigarettes/vaped after 60 minutes of waking up. Amongst conventional cigarette users, three-quarters (74.9%) smoked 10 sticks of cigarettes or less per day. Similarly, most fathers who used e-cigarettes (65.7%) reported vaping 10 times or fewer per day.

The distribution of paternal smoking by household food security categories and SES is shown in Table 3. A higher proportion of fathers residing in rural areas (48.2%) had smoking habits compared to their urban counterparts (40.9%, $p < 0.001$). Furthermore, a higher proportion of fathers who smoked were from households with lower monthly income groups (extremely low: 58.6%; low: 49.4%), whilst high-income families had the lowest proportion of paternal smokers (18.8%, $p < 0.001$). Compared with older fathers (above 49.9 years, 39.0%), a higher proportion of younger fathers smoked (below 30.0 years: 58.9%; 30.0–39.9 years: 45.7%, $p < 0.001$). A similar trend was observed in paternal education

**Table 1. Food security status and socioeconomic characteristics of children aged 0.5-12.9 years.**

| | n | Percentage | Mean | SD |
|---|---|---|---|---|
| Children's Information | | | | |
| All | 2,687 | | | |
| Age (Years) | | | 6.9 | 3.3 |
| Age Group (Years) | | | | |
| 0.5-3.9 | 593 | 22.1 | | |
| 4.0-6.9 | 819 | 30.5 | | |
| 7.0-9.9 | 658 | 24.4 | | |
| 10.0-12.9 | 617 | 23.0 | | |
| Sex | | | | |
| Boys | 1,303 | 48.5 | | |
| Girls | 1,384 | 51.5 | | |
| Area of Residence | | | | |
| Urban | 1,892 | 70.4 | | |
| Rural | 795 | 29.6 | | |
| Ethnicity | | | | |
| Malay | 1,617 | 60.2 | | |
| Chinese | 789 | 29.4 | | |
| Indian | 240 | 8.9 | | |
| Others | 41 | 1.5 | | |
| Parents' Information | | | | |
| Income (MYR) | | | 5,436 | 5,827 |
| Income Groups (MYR) | | | | |
| 2,208 and below | 659 | 24.5 | | |
| 2,209−4,849 | 847 | 31.5 | | |
| 4,850−10,959 | 947 | 35.3 | | |
| Above 10,959 | 234 | 8.7 | | |
| Paternal Age Group (Years) | | | 40.4 | 7.0 |
| Below 30.0 | 141 | 5.2 | | |
| 30.0-39.9 | 1,232 | 45.9 | | |
| 40.0-49.9 | 1,063 | 39.6 | | |
| Above 49.9 | 251 | 9.3 | | |
| Paternal Education Level | | | | |
| Non-schooling/primary education | 134 | 5.0 | | |
| Secondary education | 1,420 | 52.8 | | |
| Tertiary education | 1,133 | 42.2 | | |
| Paternal Occupation | | | | |
| Professionals/Technicians and associate professionals/Managerial | 961 | 35.8 | | |
| Clerical/ Service and sales staff | 443 | 16.5 | | |
| Agricultural, forestry and fishery/ Craft and trades/ Plant and machine/ Elementary workers | 529 | 19.6 | | |
| Armed-forces occupations | 37 | 1.4 | | |
| Self-employed/ Other occupations | 650 | 24.2 | | |
| Not working/ Retiree | 67 | 2.5 | | |
| Household Food Security | | | | |
| Household food secure | 1,627 | 60.6 | | |
| Household food insecure | 477 | 17.7 | | |
| Individual food insecure | 170 | 6.3 | | |

*(Continued)*

**Table 1.** (Continued)

| | n | Percentage | Mean | SD |
|---|---|---|---|---|
| Child hunger | 413 | 15.4 | | |

MYR represents Malaysia ringgit [USD 1 = MYR 4.13 (as at 29th November 2025)]; SD represents standard deviation.

levels, with a larger proportion of fathers with the lowest education level (59.7%) having smoking habits compared to those who completed their tertiary education (31.2%, $p < 0.001$). Fathers working as Professionals/Technicians and associate professionals/Managerial had the highest proportion of non-smokers (69.1%, $p < 0.001$) compared to Self-employed/Other occupation groups. In terms of food security, non-smoking habits were seen in a larger proportion of fathers from households with food security (62.5%). Amongst the paternal smoking groups, the highest proportion of fathers who smoked came from the households with child hunger issues (57.1%, $p < 0.001$).

Table 4 describes the OR of unadjusted and adjusted (aOR) logistic regression model of paternal smoking habits by household food security and socioeconomic categories. The odds of paternal smoking were twice as high amongst fathers aged below 30.0 years [aOR: 2.032 (95%CI 1.301–3.173)], and 1.641 times higher odds amongst fathers aged 30.0–39.9 years (95%CI 1.215–2.217), compared with fathers aged above 49.9 years. Moreover, families with very low monthly incomes had double the odds of a father who smoked [aOR: 2.606 (95%CI 1.711–3.968)] compared with high-income families. Fathers without formal education or who completed primary education had double the odds of having smoking habits [aOR:2.604 (95%CI 1.710–3.966)] compared with fathers who completed tertiary education. Fathers who worked as self-employed/ or in other occupations had 1.343 times higher odds of smoking (95%CI 1.056–1.709) compared to those who worked as professionals/technicians and associate professionals/managerial. The odds of fathers having smoking habits were also higher amongst households with food-insecure issues, [aOR: 1.251 (95%CI 1.053–1.487)] compared with the food-secure counterparts.

## Discussion

This pioneering nationwide study provided insights into underexamined links between household food insecurity and SES with paternal smoking in Malaysia. Cigarette smoking was more prevalent among younger fathers, self-employed individuals, and individuals from disadvantaged socioeconomic backgrounds, including households with lower income and fathers with lower education levels.

The present findings on the significant association between household food insecurity and smoking status are consistent with that observed in a US population study [7]. Kim-Mozeleski et al. [7] indicated a bidirectional and mutually reinforcing relationship between food insecurity and smoking status. Tobacco expenditure might result in a reciprocal effect on the food insecurity situation, while the psychological stress arising from food insecurity and the feelings of physical hunger may increase stress-induced smoking [7]. A study on young US adults aged 18–30 years revealed that the appetite suppressant effect of tobacco encouraged food-insecure individuals to smoke in order to overcome hunger [19]. Despite the impact on tobacco dependence, food insecurity is also linked with the lower likelihood of smoking cessation amongst smokers and with higher smoking initiation and relapse [7].

According to previous findings, food insecurity alone was not linked to smoking status, as individuals who experienced both food insecurity and psychological distress were significantly more likely to smoke [10]. Further research is thus warranted to identify the role of psychological distress in the relationship between food insecurity and smoking within the Malaysian context. Evidence points to the fact that individuals facing food insecurity, often associated with financial constraints and disadvantaged SES, frequently encounter more stressful environments [7]. These individuals with psychological distress have limited resources to cope with stress, which leads to higher tobacco use as a maladaptive way to relieve their psychological distress [7,20,21]. Besides, prolonged stress can further trigger smoking initiation, reducing

**Table 2. Characteristics of paternal smoking behaviours.**

| | N | Percentage |
|---|---|---|
| **Paternal Smoking** | **2,687** | |
| Smoker (Conventional & E-Cigarettes) | 1,156 | 43.0 |
| Non-smoker | 1,531 | 57.0 |
| **Cigarettes Smoking and E-cigarettes Used** | **1,156** | |
| Conventional cigarettes | 788 | 68.2 |
| E-cigarettes (ECs) | 117 | 10.1 |
| Dual users (Conventional+ECs) | 151 | 13.0 |
| Missing data | 100 | 8.7 |
| **Smoking after Waking Up (Minutes)** | **1,056** | |
| Within 5 | 148 | 14.0 |
| 6-30 | 279 | 26.4 |
| 31-60 | 164 | 15.5 |
| After 60 | 400 | 37.9 |
| Missing data | 65 | 6.2 |
| **Smoking Habit at Home** | **1,056** | |
| Not smoking/vaping at home | 383 | 36.3 |
| Smoke conventional cigarettes | 513 | 48.6 |
| Used E-cigarettes | 82 | 7.7 |
| Dual users | 53 | 5.0 |
| Missing data | 25 | 2.4 |
| **Number of Conventional Cigarettes Smoked Per Day (Sticks)** | **939** | |
| 10 and below | 703 | 74.9 |
| 11-20 | 184 | 19.6 |
| Above 20 | 4 | 0.4 |
| Missing data | 48 | 5.1 |
| **E-Cigarettes Used in Number of Sessions Per Day (ECs)** | **268** | |
| Once a day | 9 | 3.4 |
| 2–5 times | 92 | 34.3 |
| 6–10 times | 75 | 28.0 |
| 11–20 times | 36 | 13.4 |
| More than 20 times | 44 | 16.4 |
| Missing data | 12 | 4.5 |

quit-smoking attempts, and raising relapse risk [21,22]. The Theory of Planned Behaviour [23], as applied to smoking by Acarli and Kasap (2014) [24], provide a useful framework in which stress shapes positive attitudes by framing smoking as a coping strategy, thereby reinforcing the behaviour. At the same time, family disapproval can discourage smoking and promote cessation [24]. This highlights the importance of interventions that not only address stress but also strengthen family support.

This study also highlighted that amongst food-insecure families, about two out of five children (39.0% or 413 out of 1060 children) suffered from the most severe form of child hunger. Children are the most vulnerable population requiring adequate intake of nutrients for proper growth and development. Household food insecurity can negatively impact children's dietary intake, affecting the diet quality and the intake of certain food groups and nutrients [11]. A review on the effects of hunger on children described how malnutrition not only delays the development of socioemotional, cognitive, motor, and neurophysiological functioning, but also increases the risk of chronic diseases later in life. Importantly,

**Table 3. Distribution of paternal smoking status by household food security and socioeconomic characteristics categories (n = 2,687).**

| | Non-smoking | | Smoking | | Pearson's $x^2$ | p-value |
|---|---|---|---|---|---|---|
| | n | Percentage | n | Percentage | | |
| Sex | | | | | 0.012 | 0.912 |
| Boys | 741 | 56.9 | 562 | 43.1 | | |
| Girls | 790 | 57.1 | 594 | 42.9 | | |
| Age Group (Years) | | | | | 11.167 | <0.05 |
| 0.5-3.9 | 318 | 53.6 | 275 | 46.4 | | |
| 4.0-6.9 | 444 | 54.2 | 375 | 45.8 | | |
| 7.0-9.9 | 394 | 59.9 | 264 | 40.1 | | |
| 10.0-12.9 | 375 | 60.8 | 242 | 39.2 | | |
| Area of Residence | | | | | 12.236 | <0.001 |
| Urban | 1119 | 59.1 | 773 | 40.9 | | |
| Rural | 412 | 51.8 | 383 | 48.2 | | |
| Ethnicity | | | | | 79.030 | <0.001 |
| Malay | 814 | 50.3 | 803 | 49.7 | | |
| Chinese | 531 | 67.3 | 258 | 32.7 | | |
| Indian | 166 | 69.2 | 74 | 30.8 | | |
| Others | 20 | 48.8 | 21 | 51.2 | | |
| Income Group (MYR) | | | | | 177.426 | <0.001 |
| 2,208 and below | 273 | 41.4 | 386 | 58.6 | | |
| 2,209−4,849 | 429 | 50.6 | 418 | 49.4 | | |
| 4,850−10,959 | 639 | 67.5 | 308 | 32.5 | | |
| Above 10,959 | 190 | 81.2 | 44 | 18.8 | | |
| Paternal Age (Years) | | | | | 27.541 | <0.001 |
| Below 30.0 | 58 | 41.1 | 83 | 58.9 | | |
| 30.0-39.9 | 669 | 54.3 | 563 | 45.7 | | |
| 40.0-49.9 | 651 | 61.2 | 412 | 38.8 | | |
| Above 49.9 | 153 | 61.0 | 98 | 39.0 | | |
| Paternal Education Level | | | | | 114.772 | <0.001 |
| Non-schooling/primary education | 54 | 40.3 | 80 | 59.7 | | |
| Secondary education | 698 | 49.2 | 722 | 50.8 | | |
| Tertiary education | 779 | 68.8 | 354 | 31.2 | | |
| Paternal Occupation | | | | | 95.087 | <0.001 |
| Professionals/Technicians and associate professionals/Managerial | 664 | 69.1 | 297 | 30.9 | | |
| Clerical/ Service and sales staff | 233 | 52.6 | 210 | 47.4 | | |
| Agricultural, forestry and fishery/ Craft and trades/ Plant and machine/ Elementary workers | 247 | 46.7 | 282 | 53.3 | | |
| Armed-forces occupations | 16 | 43.2 | 21 | 56.8 | | |
| Self-employed/ Other occupations | 334 | 51.4 | 316 | 48.6 | | |
| Not working/ Retiree | 37 | 55.2 | 30 | 44.8 | | |
| Household Food Security | | | | | 60.851 | <0.001 |
| Household food secure | 1017 | 62.5 | 610 | 37.5 | | |
| Household food insecure | 244 | 51.2 | 233 | 48.8 | | |
| Individual food insecure | 93 | 54.7 | 77 | 45.3 | | |
| Child hunger | 177 | 42.9 | 236 | 57.1 | | |

MYR represents Malaysia ringgit [USD 1 = MYR 4.13 (as at 29[th] November 2025)].

p-value indicates the significance level by using Pearson's Chi-Square test.

**Table 4. Odds ratio for paternal smoking status by household food security and socioeconomic characteristics categories (n = 2,687).**

| | Model 1 | | | | Model 2 | | | |
|---|---|---|---|---|---|---|---|---|
| | OR | 95%CI | Cox & Snell R square | Nagelkerke R square | aOR | 95%CI | Cox & Snell R square | Nagelkerke R square |
| | | | 0.087 | 0.117 | | | 0.1.00 | 0.134 |
| Income[a] (MYR) | | | | | | | | |
| 2,208 and below | 3.298*** | 2.191-4.964 | | | 2.606*** | 1.711-3.968 | | |
| 2,209−4,849 | 2.696*** | 1.841-3.949 | | | 2.380*** | 1.618-3.501 | | |
| 4,850−10,959 | 1.684** | 1.171-2.421 | | | 1.627** | 1.129-2.344 | | |
| Above 10,959 | 1 | | | | 1 | | | |
| Paternal Education Level[b] | | | | | | | | |
| Non-schooling/primary education | 2.007** | 1.336-3.016 | | | 2.604*** | 1.710-3.966 | | |
| Secondary education | 1.450*** | 1.186-1.771 | | | 1.585*** | 1.292-1.945 | | |
| Tertiary education | 1 | | | | 1 | | | |
| Paternal Occupation[c] | | | | | | | | |
| Clerical/ Service and sales staff | 1.199 | 0.928-1.549 | | | 1.199 | 0.926-1.553 | | |
| Agricultural, forestry and fishery/ Craft and trades/ Plant and machine/ Elementary workers | 1.218 | 0.933-1.590 | | | 1.219 | 0.931-1.596 | | |
| Armed-forces occupations | 1.706 | 0.848-3.433 | | | 1.468 | 0.729-2.955 | | |
| Self-employed/ Other occupations | 1.308* | 1.032-1.658 | | | 1.343* | 1.056-1.709 | | |
| Not working/ Retiree | 1.136 | 0.659-1.959 | | | 1.069 | 0.620-1.844 | | |
| Professionals/Technicians and associate professionals/Managerial | 1 | | | | 1 | | | |
| Paternal Age[d] (Years) | | | | | | | | |
| Below 30.0 | 2.084** | 1.337-3.248 | | | 2.032** | 1.301-3.173 | | |
| 30.0-39.9 | 1.675** | 1.241-2.259 | | | 1.641** | 1.215-2.217 | | |
| 40.0-49.9 | 1.207 | 0.894-1.630 | | | 1.237 | 0.915-1.673 | | |
| Above 49.9 | 1 | | | | 1 | | | |
| Household Food Security[e] | | | | | | | | |
| Food insecure | 1.289** | 1.086-1.530 | | | 1.251* | 1.053-1.487 | | |
| Food secure | 1 | | | | 1 | | | |

95% CI represents 95% confidence interval; aOR represents adjusted odds ratio; MYR represents Malaysia ringgit [USD 1 = MYR 4.13 (as at 29th November 2025)]; OR represent odds ratio.

Model 1: unadjusted model; Model 2: adjusted with ethnicity and area of residence.

Significant odds ratio using binary logistic regression at *$p < 0.05$, **$p < 0.01$ and ***$p < 0.001$.

Reference category of paternal smoking is non-smoking.

[a] Reference category of income group is above MYR 10959.

[b] Reference category of paternal education level is tertiary education level.

[c] Reference category of paternal occupation is professional/technician/managerial category.

[d] Reference category of paternal age group is above 49.9 years old.

[e] Reference category of household food security is food secure; food insecure includes household food insecure, individual food insecure, and child hunger.

child hunger is associated with depression and suicidal thoughts during late adolescence and adulthood [25]. Given that paternal smoking may exacerbate financial strain, intensify household stress, and influence food allocation within already food-insecure households, further research is warranted to explore the relationship between paternal smoking and children's dietary patterns, nutritional status, and overall well-being development.

In accordance with previous studies [10,26–29], individuals with lower SES, household income, and education levels are associated with increased susceptibility to smoking habits. Being a smoker may contribute to nutritional deprivation in a community which is already disproportionately affected by poverty and poor health outcomes. Persistent exposure to a financially unstable environment can influence behaviour and decision-making. When financial resources are perceived to be limited, they might spend less on food while increasing their expenditure on tobacco, leading to purchase of more processed foods that consequently result in unhealthy eating habits [26,30,31]. Allocating household income to tobacco use instead of nutrition, education, and healthcare can also negatively impact the physical and mental development of children. Financial constraints had also been reported to directly induce limited access to cognitively stimulating materials and preschool experiences [32,33]. Furthermore, economic challenges have been associated with reduced intention of smoking cessation [27], and the situation may further worsen if individuals suffer from a combination of economic difficulties and food insecurity [10]. A cohort study also revealed that smokers facing greater socio-economic disadvantages exhibited lower receptiveness to smoking-cessation treatments like pharmacotherapy or behavioural strategies [34].

As indicated in our study, fathers with higher levels of education were more inclined to refrain from smoking, possibly due to their enhanced awareness of health risks and perceived severity of smoking-related harms [35]. In line with the Health Belief Model, greater awareness may enhance individuals' perception of severity and susceptibility to non-communicable diseases, thereby motivating the avoidance of harmful behaviours, including smoking [36]. In this context, pictorial warnings on tobacco packaging function as an important health communication tool. Local evidence suggests they are particularly effective among non-smokers, who may already hold stronger risk perceptions and therefore respond more sensitively to visual cues [37]. Consequently, highly educated populations are more likely to adopt healthier lifestyles, engage in improved self-care, respond more effectively to health warnings, and experience a reduction in exposure to hazardous environments. This leads to longer life expectancy and greater satisfaction [38]. Moreover, better self-coping skills enable them to alleviate stress without resorting to tobacco use [22]. A survey conducted among the Japanese population revealed that education influences habits of early life, while occupation plays a role in determining behaviour in later stages [39]. Education may profoundly affect smoking initiation among the younger population, whereas occupation has a greater impact on smoking cessation, continuation, or relapse among middle-aged or older populations [39]. Therefore, individuals with higher levels of education may enjoy better employment opportunities, potentially leading to a reduced urge to start smoking throughout their lives.

Furthermore, our results showed that self-employed fathers had a higher likelihood of smoking. A local publication had suggested that self-employed groups might suffer from financial instability and grapple with numerous compliance and regulatory burdens, potentially leading to increased stressors [40]. The present preliminary analysis also found that most of the self-employed families (66.5%) were grouped as extremely low- or low-income (presented in S1 Table). Therefore, these fathers might be more likely to have stress-induced smoking habits.

Being younger remains a strong indicator of a propensity to smoke, aligning with findings from previous studies [28,41,42]. The onset of health problems in older populations could encourage cessation of unhealthy habits like tobacco use [43]. Additionally, older individuals may have greater self-awareness regarding smoking cessation due to frequent exposure to anti-smoking messages in health facilities [44]. Another plausible explanation was related to the SES conditions, as older individuals may be more economically and financially stable, reducing the need for tobacco use [34].

Consistent with a local study [28], the current results of tobacco addiction among smokers demonstrated a relatively low nicotine dependence, as most conventional smokers reported consuming 10 sticks of cigarettes or less per day or/ and vaped 10 times or less per day. This observation highlights an opportunity for policymakers and stakeholders to invest resources in targeted smoking cessation programmes, such as expanding access to quit-smoking clinics – particularly for disadvantaged populations, alongside reducing illicit cigarette availability, thereby capitalising on the higher chance of quitting when nicotine dependence is low. To further curb smoking dependency, strict taxation measures represent a promising long-term strategy, as highlighted by local evidence [45]. However, tobacco tax currently accounts for only 58.6% of the retail price in Malaysia [46], falling short of the WHO-recommended threshold of 75% [47]. A revision of taxation

policies is therefore warranted. Furthermore, given the increasing trend of e-cigarette use, the implementation of robust tax enforcement alongside pictorial health warnings should be extended to e-cigarettes to counter their rising popularity. This study also drew attention to children's passive smoking exposure, as more than 60% of smoking fathers claimed they smoked at home. Children exposed to passive smoking face increased risks of morbidity and mortality, as well as a higher likelihood of initiating smoking early and continuing it until adulthood [48,49]. This could pose a challenge to establishing a smoke-free environment for future generations.

Employing a nationwide sample in this present study provides valuable insights into paternal smoking habits among the Malaysian population and their associated factors. This exploratory study in Malaysia attempted to link household food security with smoking habits, offering crucial insights, especially for future studies examining the impact of paternal smoking on children's dietary intake and nutritional status, particularly in low-income families. Besides, incorporating the identified confounders enhances the accuracy of capturing contributing factors. However, unmeasured confounders such as psychological influences may still exist and should be considered in future studies. The current study did not apply complex survey procedures in the analysis, which may have influenced the estimated standard errors and statistical significance. As such, the findings should be interpreted with appropriate caution. Although the cross-sectional design of this study was unable to confirm a causal relationship between the factors studied and paternal smoking, it provides important groundwork for future longitudinal exploration to uncover deeper links between social determinants of health, household food security, and paternal smoking.

## Conclusions

The study shows that Malaysian fathers of children aged 0.5–12.9 years from households experiencing food insecurity and lower income, along with lower education levels, younger age, and self-employment, were more likely to smoke. Addressing these socioeconomic barriers and ensuring equitable access to smoking cessation resources remain crucial. Strengthening efforts to combat food insecurity and socioeconomic disparities are essential, given their close links with smoking prevalence.

To create a healthier environment for the next generation, smoking cessation programmes should be prioritised, as this study found that addiction levels remain relatively low, making quitting more achievable. The proposed 13th Malaysia Plan, announced in 2025, underscores pro-health taxation, with levies on products that are harmful to health, and its expansion to tobacco and vaping products, is a critical step to drive behavioural change and raise health programmes funding. Complementary actions, such as further increasing tobacco taxes or phasing out vaping products, are also strongly recommended. Equally important, there is a need for sustainable public health programmes designed to monitor and address the effects of SHS and THS smoke on children, particularly given that 60% of fathers who smoke reported smoking at home. Taken together, these measures could significantly contribute to achieving the United Nations Sustainable Development Goals, specifically targeting Goal 2: Zero Hunger and Goal 3: Good Health and Well-being for future generations.

## Supporting information

**S1 Table. Household income among self-employed fathers (n = 645).**
(DOCX)

## Acknowledgments

The authors thank all the subjects, parents, and caregivers for their full cooperation, as well as teachers, community leaders, and all involved in the study for their support during the conduct of this study. The researchers, data collection team, enumerators, and all those involved in this project are acknowledged and appreciated for their dedication and effort.

The SEANUTS II Malaysia Study Group comprises the following. Universiti Kebangsaan Malaysia: Bee Koon Poh (lead), Jyh Eiin Wong, Nik Shanita Safii, Mohamad Fauzi Nor Farah, Nor Aini Jamil, Razinah Sharif, Caryn Mei Hsien Chan, Swee Fong Tang, Lei Hum Wee, Siti Balkis Budin, Denise Koh, Abd Talib Ruzita, Nur Zakiah Mohd Saat, Mohd Jamil Sameeha, A. Karim Norimah, See Meng Lim, Jasmine Siew Min Chia, Shoo Thien Lee. FrieslandCampina: Ilse Khouw, Swee Ai Ng, Nanda de Groot.

During the preparation of this work, the authors used ChatGPT for English proofreading. After using this tool/service, authors reviewed and edited the content as needed and take full responsibility for the content of the publication.

## Author contributions

**Conceptualization:** Lei Hum Wee, Bee Koon Poh.

**Data curation:** Giin Shang Yeo, Shoo Thien Lee, Kuan Chiet Teh.

**Formal analysis:** Giin Shang Yeo, Shoo Thien Lee, Kuan Chiet Teh.

**Funding acquisition:** Bee Koon Poh.

**Methodology:** Giin Shang Yeo, Kuan Chiet Teh, Lei Hum Wee, Bee Koon Poh.

**Project administration:** Jyh Eiin Wong, Bee Koon Poh.

**Supervision:** Jyh Eiin Wong, Caryn Mei Hsien Chan, Nur Zakiah Mohd Saat, Nik Shanita Safii, Siti Balkis Budin, Swee Fong Tang, A. Karim Norimah, Lei Hum Wee, Bee Koon Poh, on behalf of the SEANUTS II Malaysia Study Group.

**Validation:** Shoo Thien Lee.

**Visualization:** Giin Shang Yeo, Shoo Thien Lee.

**Writing – original draft:** Giin Shang Yeo.

**Writing – review & editing:** Giin Shang Yeo, Shoo Thien Lee, Kuan Chiet Teh, Jyh Eiin Wong, Wan Siti Fatimah Wan Abdul Rahman, Nurul Hasanah Hasmuni Chew, Caryn Mei Hsien Chan, Nur Zakiah Mohd Saat, Nik Shanita Safii, Siti Balkis Budin, Swee Fong Tang, A. Karim Norimah, Ilse Khouw, Lei Hum Wee, Bee Koon Poh.

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
