## [Decision Letter · Decision Letter 0]

22 Aug 2025

Dear Dr. Poh,

We look forward to receiving your revised manuscript.

Kind regards,

Seo Ah Hong, PhD

Academic Editor

PLOS ONE

Journal Requirements:

“This study was conducted as part of the South East Asian Nutrition Surveys II (SEANUTS II) (NN-2018-159) and was funded by FrieslandCampina, Amersfoort, The Netherlands. FrieslandCampina was not involved in the recruitment of children and the final set of results.”

3. Please note that funding information should not appear in the Acknowledgments section or other areas of your manuscript. We will only publish funding information present in the Funding Statement section of the online submission form. Please remove any funding-related text from the manuscript.

“Ilse Khouw is an employee of FrieslandCampina. All other authors declare no competing interest.”

We note that one or more of the authors are employed by a commercial company: FrieslandCampina

2) Please also provide an updated Competing Interests Statement declaring this commercial affiliation along with any other relevant declarations relating to employment, consultancy, patents, products in development, or marketed products, etc.

Within your Competing Interests Statement, please confirm that this commercial affiliation does not alter your adherence to all PLOS ONE policies on sharing data and materials by including the following statement: ""This does not alter our adherence to PLOS ONE policies on sharing data and materials.” (as detailed online in our guide for authors http://journals.plos.org/plosone/s/competing-interests) . If this adherence statement is not accurate and there are restrictions on sharing of data and/or materials, please state these. Please note that we cannot proceed with consideration of your article until this information has been declared.

5. We note that you have indicated that there are restrictions to data sharing for this study. For studies involving human research participant data or other sensitive data, we encourage authors to share de-identified or anonymized data. However, when data cannot be publicly shared for ethical reasons, we allow authors to make their data sets available upon request. For information on unacceptable data access restrictions, please see http://journals.plos.org/plosone/s/data-availability#loc-unacceptable-data-access-restrictions.

6. In this instance it seems there may be acceptable restrictions in place that prevent the public sharing of your minimal data. However, in line with our goal of ensuring long-term data availability to all interested researchers, PLOS’ Data Policy states that authors cannot be the sole named individuals responsible for ensuring data access (http://journals.plos.org/plosone/s/data-availability#loc-acceptable-data-sharing-methods).

7. When completing the data availability statement of the submission form, you indicated that you will make your data available on acceptance. We strongly recommend all authors decide on a data sharing plan before acceptance, as the process can be lengthy and hold up publication timelines. Please note that, though access restrictions are acceptable now, your entire data will need to be made freely accessible if your manuscript is accepted for publication. This policy applies to all data except where public deposition would breach compliance with the protocol approved by your research ethics board. If you are unable to adhere to our open data policy, please kindly revise your statement to explain your reasoning and we will seek the editor's input on an exemption. Please be assured that, once you have provided your new statement, the assessment of your exemption will not hold up the peer review process.

8. One of the noted authors is a group or consortium: SEANUTS II Malaysia Study Group

In addition to naming the author group, please list the individual authors and affiliations within this group in the acknowledgments section of your manuscript. Please also indicate clearly a lead author for this group along with a contact email address.

9. We note that you have included the phrase “data not shown” in your manuscript. Unfortunately, this does not meet our data sharing requirements. PLOS does not permit references to inaccessible data. We require that authors provide all relevant data within the paper, Supporting Information files, or in an acceptable, public repository. Please add a citation to support this phrase or upload the data that corresponds with these findings to a stable repository (such as Figshare or Dryad) and provide and URLs, DOIs, or accession numbers that may be used to access these data. Or, if the data are not a core part of the research being presented in your study, we ask that you remove the phrase that refers to these data.

**Additional Editor Comments:**

1. Please revise the title to clarify. Does this refer to the association between household food security, socioeconomic status, and paternal smoking? Since this was a cross-sectional study, "association" should be used rather than "relationship."

2. The dependent variable of this study is smoking, and the main independent variables are household food security and socioeconomic status. The introduction should summarize an intensive review of the literature on the relationships between these variables and describe the research gaps; however, this is missing. While the discussion section provides information on existing literature, the discussion on what research gaps exist and how this study addresses them remains weak. The study suggests that smoking is prevalent among socioeconomically disadvantaged (food insecure) groups. What new information does this study provide compared to the existing literature? Furthermore, the introduction described conventional cigarette smoking and e-cigarette use, which I expected to assess the relationship between these smoking patterns and household food security and socioeconomic status. However, this analysis was not included, and instead, smoking patterns were presented only descriptively. The revision should focus more on what interesting insights this study offers to international readers compared to existing knowledge.

Reviewers' comments:

Reviewer's Responses to Questions

**Comments to the Author**

1. Is the manuscript technically sound, and do the data support the conclusions?

Reviewer #1: Yes

Reviewer #2: Partly

2. Has the statistical analysis been performed appropriately and rigorously?

Reviewer #1: Yes

Reviewer #2: No

3. Have the authors made all data underlying the findings in their manuscript fully available?

Reviewer #1: Yes

Reviewer #2: No

4. Is the manuscript presented in an intelligible fashion and written in standard English?

Reviewer #1: Yes

Reviewer #2: Yes

Reviewer #1: Congratulations to the author for contributing to a better understanding of this important public health concern—paternal smoking. Below are some suggestions for your consideration. Please take action on any comments you find relevant.

Thank you.

Title

• The title is clear and appropriately reflects the study's focus.

Abstract

• [Lines 21-23] The introduction does not clearly articulate the need for the present study. Consider explicitly stating the research gap or rationale.

• [Line 24] There is a slight inconsistency in the recruitment description. While the study recruited children, the analysis focuses on paternal smoking as reported by parents. This could lead to misinterpretation regarding the source of the data. Clarifying this would improve accuracy.

• [Lines 24-26] If the sample is derived from SEANUTS II Malaysia, the sentence should be revised for clarity to ensure accurate representation of the data source.

• [Lines 26-27] It would be beneficial to explicitly mention that information on paternal smoking was gathered through parent/guardian-reported questionnaires.

Introduction

• [Overall] Consider adding a transitional sentence to enhance coherence between ideas.

• [Overall] The introduction highlights the high prevalence of paternal smoking and household food insecurity but does not explicitly state the gap in existing research or the necessity of this study. Strengthening this aspect would improve justification for the research.

Methods

• [Lines 80-82] Instead of solely referring to previous articles, it would be helpful to briefly summarize key aspects of the study design to provide context.

• [Line 90] Elaborating on how paternal smoking status was determined—whether based on self-reported frequency, duration, or another criterion—would improve clarity.

• [Line 118] "Pearson’s" should be revised for grammatical correctness.

• [Line 119] Consider using “paternal smoking status” instead of “paternal smoking patterns” to accurately reflect the binary classification in the logistic regression analysis.

• [Line 122] Clarifying the rationale for adjusting Model 2 with “ethnicity” and “area of residence” would strengthen the methodological justification. Why were these particular variables selected for control?

Results

• [Line 123] The study does not include socioeconomic status (SES) data for mothers, despite maternal employment, education, and income being key contributors to household SES. Providing a justification for this omission or discussing its potential impact on the findings would enhance the study’s clarity.

• [Line 158] What were the selection criteria for variables included in the logistic regression analysis? Clarifying this would enhance the methodological transparency.

Discussion

• [Lines 175-177] The discussion references associations between dietary intake and children's physiological development. Are these findings within the scope of the current study? If not, consider refining the discussion to align more closely with the study's objectives.

• [Lines 195-197] Since psychological factors were not included as study variables, it would be more appropriate to suggest that psychological distress might be a missing link and recommend further research to explore this relationship. Proposing its inclusion in interventions may be premature without direct evidence from the study, particularly in the Malaysian context.

• [Lines 187-207] The paragraph structure needs reorganization. Prioritize explaining the study's significant findings and their implications. Recommendations regarding psychological factors and dietary intake should either be placed later in the discussion or moved to the conclusion.

• [Lines 248-251] The recommendation for targeted smoking cessation programs should be incorporated into the conclusion rather than the discussion section.

Conclusion

• [Overall] Some recommendations currently presented in the discussion would be more appropriately included here as future directions for addressing smoking prevalence. Consider summarizing key findings and suggesting evidence-based interventions.

Reviewer #2: Thank you for inviting me to review the manuscript, title: "Relationship between household food security and socioeconomic status with paternal smoking: Findings from SEANUTS II Malaysia". Manuscript Number: PONE-D-25-11571

⚠� 1. Study Design and Methodology ⚠

Strengths - The nationwide cross-sectional design and large sample size (n=2,687) enhance generalizability. Ethical approvals and multi-stage cluster sampling are clearly reported.

I want to point out that there are 3 majors issue need to improved in study design and methodology:

i) Missing Definition of Dependent Variable (Paternal Smoking Status) in Method section

While the study provides valuable insights into the association between household food insecurity, socioeconomic status, and paternal smoking in Malaysia, a significant methodological omission is the lack of a clear operational definition for the dependent variable — paternal smoking status — in the Methods section. Although Table 2 distinguishes between conventional cigarette use, e-cigarette use, and dual use, it remains unclear how these categories were collapsed into the binary variable used in the logistic regression model. Specifically, it is not stated whether:

- Only current smokers were included in the "smoker" category,

- Former or occasional smokers were excluded or grouped with non-smokers,

- Any minimum threshold of frequency or duration of smoking was applied.

The binary smoker/non-smoker designation does not account for type (e.g., exclusive vs dual use), intensity (e.g., number of cigarettes/day), duration, or cessation history. Furthermore, data on household smoking behavior (e.g., smoking at home) is described but not utilized in analysis, despite being highly relevant to child exposure. This information is essential for replicability and for interpreting the adjusted odds ratios reported in Table 4. Additionally, the authors note in the Results that "the OR describes the probability of children’s fathers without smoking habits (reference group) relative to those with smoking habits," yet this statement lacks sufficient detail to evaluate the validity of the categorization.

My recommendation:

The Methods section should be revised to include a clear, unambiguous definition of paternal smoking status, including:

- The specific survey question(s) used to assess smoking behavior,

- Criteria used to categorize participants as smokers or non-smokers (e.g., current use of conventional cigarettes, e-cigarettes, or both),

- How occasional, former, or missing responses were handled.

Clarifying this critical aspect of the study design will substantially strengthen the methodological transparency and reproducibility of the manuscript.

ii) Absence of control for potential psychological variables like stress or mental health in adjusted models / Omitted Confounders

Additionally, while the manuscript extensively discusses psychological stress and coping behavior as drivers of smoking, no mental health variables are included as confounders in the logistic regression models. This raises concern about potential omitted variable bias in the association between socioeconomic status and smoking. The authors should either incorporate these critical variables (if available) or explicitly discuss the limitation.

iii) No mention of cluster-adjusted standard errors despite multistage cluster sampling

The manuscript describes the use of a multistage cluster sampling strategy to recruit participants, which is appropriate for a nationwide survey. However, no mention is made of whether the statistical analyses accounted for the clustered nature of the data. Ignoring clustering can lead to underestimated standard errors and inflated Type I error rates, potentially compromising the validity of the reported associations. The authors should clarify whether cluster-adjusted standard errors or complex survey procedures were used in the logistic regression models. If not, this limitation should be explicitly acknowledged and discussed.

2. Results and Data Analysis

Strengths - Results are statistically robust, with clear presentation of unadjusted and adjusted logistic regression. Tables are well-structured and easy to interpret.

Critiques & Concerns

a. No multicollinearity check reported for logistic regression model

b. No justification for included covariates in Model 2 (why only ethnicity and area?)

c. Smoking behavior categories (e.g., home smoking, first smoke after waking) are described but not statistically analyzed

d. Lack of reporting on missing data handling strategy

Recommendations

a. Report variance inflation factors (VIF) or other diagnostics to confirm model validity.

b. Provide rationale for covariate selection, or consider including more relevant confounders like child's age.

c. Consider additional analyses (e.g., multinomial regression or stratified models) to explore nuanced behavioral associations.

d. Detail any imputation method used or clarify if complete case analysis was performed.

3. Writing Quality and Structure

Strengths: Well-written, clear academic language. Structured logically. Abstract summarizes key findings effectively.

Critiques & Concerns : Discussion sometimes reiterates results without deeper interpretation

Recommendations : Emphasize mechanisms, policy implications, and compare more with international findings.

4. Literature Review and Citations

Strengths: The references are current, relevant, and diverse in origin (including Malaysian and international sources). Well-anchored in public health literature.

Critiques & Concerns

a. Limited discussion of literature from Southeast Asia (other than Malaysia)

b. Some important psychological literature under-cited (stress-smoking pathways)

Recommendations

a. Consider referencing regional studies from Indonesia, Philippines, or Thailand to contextualize findings.

b. Include more theoretical models (e.g., Health Belief Model, Theory of Planned Behavior) to deepen analysis.

5. Figures and Tables

Strengths: Tables are comprehensive and well-labeled. Statistical details (OR, CI, p-values) are clearly provided.

Critiques & Concerns

a. Table 3: No adjusted percentages for smoking prevalence across SES

b. No figures presented

Recommendations

a. Consider reporting adjusted percentages or use logistic regression margins to show practical effect sizes.

b. Consider including a figure (e.g., forest plot of ORs or a conceptual diagram of model pathways).

6. Ethical Considerations

Strengths: Ethical approval (UKM JEP-2018-569) and informed consent procedures are clearly documented. Trial registration is provided.

Critiques & Concerns: Data availability is restricted

Recommendations : Consider sharing de-identified or synthetic data or providing an open metadata description.

7. Novelty and Significance

Strengths: Novel exploration of paternal smoking in relation to household food insecurity in Malaysia. Addresses a relevant and urgent public health issue.

Critiques & Concerns

a. Causality cannot be inferred (e.g., does food insecurity lead to smoking, or vice versa?)

b. Limited discussion of practical implications beyond general calls for intervention

Recommendations

a. Emphasize this limitation and recommend longitudinal research.

b. Include more specific public health policy suggestions (e.g., integration into tobacco tax reallocation).

Most Critical Issues (Prioritized):

- Lack of clear definition of the dependent variable (paternal smoking status) in the Methods section

- Failure to account for multistage cluster sampling in statistical analysis

- Omission of psychological or mental health variables as confounders

- Oversimplification of smoking behavior (e.g., intensity, frequency, cessation)

- Disconnect between stated risks (secondhand/thirdhand smoke) and variables analyzed

- Lack of justification for covariate selection in adjusted regression models

- Missing data and exclusion not properly addressed

Final Recommendation: Major Revision

**Do you want your identity to be public for this peer review?** For information about this choice, including consent withdrawal, please see our Privacy Policy

Reviewer #1: No

Reviewer #2: **Yes:**  Win Khaing

---

## [Author Response · Author response to Decision Letter 1]

24 Oct 2025

We would like to thank the Editor and reviewers for their constructive comments, which have greatly contributed to improving our manuscript. Detailed responses to all comments have been provided in a separate file titled ‘Responses to Reviewers.

---

## [Decision Letter · Decision Letter 1]

13 Nov 2025

Dear Dr. Poh,

We look forward to receiving your revised manuscript.

Kind regards,

Seo Ah Hong, PhD

Academic Editor

PLOS ONE

Journal Requirements:

Reviewers' comments:

Reviewer's Responses to Questions

**Comments to the Author**

Reviewer #1: All comments have been addressed

Reviewer #2: All comments have been addressed

2. Is the manuscript technically sound, and do the data support the conclusions?

Reviewer #1: Yes

Reviewer #2: Yes

3. Has the statistical analysis been performed appropriately and rigorously?

Reviewer #1: Yes

Reviewer #2: Yes

4. Have the authors made all data underlying the findings in their manuscript fully available?

Reviewer #1: Yes

Reviewer #2: Yes

5. Is the manuscript presented in an intelligible fashion and written in standard English?

Reviewer #1: Yes

Reviewer #2: Yes

Reviewer #1: All previous comments have been adequately addressed.

There are a few additional recommendations below, which the journal or authors may consider for further improvement. These suggestions are optional.

• [l134–135] It is suggested to remove “household food security levels” and “paternal smoking status” from this section, as the mean and standard deviation values for these variables are not reported in the Results section.

• [l259–260] The transition towards recommending further study on the association between smoking habits and children’s diet, body weight status, and health appears abrupt and somewhat disconnected from the earlier content in this paragraph, which focused on consequences of household food insecurity. It is recommended to provide additional explanation or linkage to strengthen the rationale for this recommendation.

Reviewer #2: Overall Assessment

I sincerely appreciate the authors for their thorough and thoughtful revisions. The revised manuscript has significantly improved in clarity, methodological rigor, and interpretability. The authors have addressed all of my major and minor concerns with commendable diligence and transparency.

Response to Previous Comments

Paternal Smoking Status

The authors have successfully incorporated the requested clarifications and adjustments related to paternal smoking status. The revised analytical framework now reflects a more accurate and comprehensive assessment of exposure. This revision strengthens both the internal validity and interpretive depth of the findings.

Psychological Variables (Stress and Mental Health)

The concern regarding the potential influence of psychological variables such as stress or mental health on the outcomes has been thoughtfully acknowledged in the revised limitations section. Although direct measurement of these factors was not feasible, the authors have appropriately discussed their potential confounding effect and justified their absence in the adjusted models.

Cluster-Adjusted Analysis

The issue regarding clustering or potential intra-group correlation has also been addressed. The authors clearly stated the rationale for their analytical approach and acknowledged this aspect as a limitation. Their discussion reflects a sound understanding of the underlying methodological implications.

Minor Revisions and Overall Quality

The manuscript’s language, structure, and presentation have improved markedly. The tables and figures are well-organized, and the discussion section demonstrates greater coherence between empirical findings and theoretical interpretation. The response letter also provides transparent explanations of each revision, which reflects academic rigor and integrity.

Conclusion

In light of the authors’ careful revisions and comprehensive responses to all major and minor comments, I am satisfied that the manuscript now meets the standards required for publication. The study offers valuable insights and will contribute meaningfully to the literature in this area.

Recommendation: Accept for publication.

**Do you want your identity to be public for this peer review?** For information about this choice, including consent withdrawal, please see our Privacy Policy

Reviewer #1: No

Reviewer #2: **Yes:**  Win Khaing

---

## [Author Response · Author response to Decision Letter 2]

19 Nov 2025

We sincerely thank the reviewers for their time and for providing constructive comments that have greatly strengthened the quality of our manuscript.

---

## [Editor Report · Decision Letter 2]

26 Nov 2025

Association between household food security and socioeconomic status with paternal smoking: Findings from SEANUTS II Malaysia

PONE-D-25-11571R2

Dear Dr. Poh,

We’re pleased to inform you that your manuscript has been judged scientifically suitable for publication and will be formally accepted for publication once it meets all outstanding technical requirements.

Kind regards,

Seo Ah Hong, PhD

Academic Editor

PLOS ONE
---

## [Editor Report · Acceptance letter]

PONE-D-25-11571R2

PLOS One

Dear Dr. Poh,

I'm pleased to inform you that your manuscript has been deemed suitable for publication in PLOS One. Congratulations! Your manuscript is now being handed over to our production team.

Kind regards,

on behalf of

Prof. Seo Ah Hong

Academic Editor

PLOS One